# Characteristics of Synovial Fistula of the Ankle Joint: A Case Series

**DOI:** 10.3390/jcm11206215

**Published:** 2022-10-21

**Authors:** Chul-Hyun Park, Jeong Jin Park, In Ha Woo, Hongfei Yan, Woo-Chun Lee

**Affiliations:** 1Department of Orthopaedic Surgery, College of Medicine, Yeungnam University, Daegu 42415, Korea; 2Department of Orthopaedic Surgery, Yeungnam University Medical Center, Daegu 42415, Korea; 3Seoul Foot and Ankle Center, Dubalo Orthopaedic Clinic, Seoul 04551, Korea

**Keywords:** ankle, synovial fistulae, characteristics, prognosis

## Abstract

(1) Background: Little is known about the etiology, clinical features, diagnosis methods, treatments, and the prognosis of synovial fistula of the ankle joint. The purpose of this study is to investigate the clinical features of synovial fistula of the ankle joint. (2) Methods: Between March 2003 and December 2018, 40 cysts associated with synovial fistula of the ankle joint were treated consecutively by two surgeons. Case histories, clinical manifestations, intraoperative findings, surgical treatment methods, and treatment outcomes were evaluated to characterize fistula-associated cysts. The clinical results were assessed using the visual analog scale (VAS) and the American Orthopaedic Foot and Ankle Society (AOFAS) ankle-hindfoot functional scores, preoperatively and at the last follow-up. (3) Results: The main complaints were ankle instability and pain (15 patients), pain only (15 patients), instability (seven patients), and cosmetic problems (three patients). Eleven patients had a cyst with an open skin wound, and eight of these patients had undergone surgery under a misdiagnosis of bursitis. Cysts were located anterior to the lateral malleolus in 22 cases, next to the lateral malleolus in 13 cases, posterior to the lateral malleolus in three cases, and across the entire lateral malleolus in two cases. Mean VAS and AOFAS scores improved from 5.2 (range, from 1 to 7) and 72.3 (range, from 65 to 87) preoperatively to 1.1 (range, from 0 to 3) and 93.6 (range, from 85 to 100), respectively, at final follow-up visits. (4) Conclusions: Cyst occurrence due to a synovial fistula should be considered when treating a cyst around the lateral malleolus. Fistula repair and reinforcement with fibular periosteum provides a good treatment option for cysts attributed to synovial fistula of the ankle that fail to respond to conservative treatment.

## 1. Introduction

A fistula is a channel that connects two body cavities or breaks through a normal tissue barrier [1]. A synovial fistula is a condition caused by a passage or communication between a synovial space and a joint. The condition generally occurs after arthrotomy or a trauma-associated joint capsule rupture in the knee joint [2,3,4,5,6,7], though it has been reported in the shoulder or arm [8,9,10,11,12,13]. A synovial fistula can also be caused by complications of rheumatoid arthritis and by suture material reactions [14,15]. However, due to the low incidence of synovial fistula, its clinical manifestations, diagnostic methods, and treatment are not well known.

Cysts that occur around the ankle joint are often caused by bursitis [16]. However, for cysts that occur after ankle sprains, the possibility of a synovial fistula caused by a rupture of the joint capsule should not be excluded because if a cyst caused by a synovial fistula is misdiagnosed as bursitis, treatment can fail due to the occurrence of a chronic wound or infection after surgical treatment. Understanding the characteristics of synovial fistula of the ankle is required, but comparatively little is known about the characteristics of this condition.

We have experienced several cysts caused by synovial fistula of the ankle joint after ankle sprains. This study was undertaken to evaluate its clinical manifestations, diagnosis, treatment methods, and treatment outcomes.

## 2. Materials and Methods

This study was approved by the institutional review board of our hospital, which waived the requirement for informed consent because of its retrospective design. Between March 2003 and December 2018, 40 cysts (40 patients) associated with synovial fistula of the ankle joint were treated consecutively by two surgeons using the same surgical methods. All patients were available for a complete follow-up at an average of 55.8 months (24–156). The subjects were 19 males and 21 females with a mean age of 54.7 years (19–79). All patients who were included had a magnetic resonance imaging (MRI)- or arthrography-confirmed cyst to ankle joint communication. Initially, patients underwent short leg splint immobilization for 2 weeks and surgical treatment after conservative treatment failures.

Case histories, clinical manifestations, intraoperative findings, surgical treatment methods, and treatment outcomes were evaluated to characterize fistula-associated cysts. The clinical results were assessed using a visual analog scale (VAS) and the American Orthopedic Foot and Ankle Society (AOFAS) ankle-hindfoot functional scores, preoperatively and at final follow-up [17]. To prevent potential bias, clinical evaluations were performed by an independent observer who was not part of the surgical team.

### 2.1. Surgical Technique

The procedure was performed with patients in the lateral decubitus position under general or spinal anesthesia. Initially, the affected leg was exsanguinated with an elastic bandage, and a tourniquet was applied to the thigh. A 5-cm curvilinear incision was then made along the anterior margin of the lateral malleolus, avoiding the superficial peroneal nerve. After locating the fistula capsule around the cyst, synovial tissues around the fistula were debrided. Saline was loaded into the joint, and saline leakage was checked when the opening was difficult to find (Figure 1). Fistulae were sutured using absorbable sutures.

After fistula repair, when soft tissues around the suture site were poor or lacking, augmentation was performed using a fibular periosteal or peroneal fascia flap, as previously described [18,19]. Flaps were secured by mattress sutures to repaired sites using an absorbable suture.

In cases with accompanying ankle instability, additional surgery such as the modified Brostrom procedure [20], the Chrisman–Snook procedure [21], or anatomical ligament reconstruction [22] were performed, according to the degree of instability. In cases of severe soft tissue defect, a reverse sural artery flap was used.

### 2.2. Postoperative Care

A short leg cast was applied in the neutral position for four weeks after surgery, followed by an air-stirrup brace for another 2 months. Partial weight bearing was permitted with a cast from two weeks after surgery. At 4 weeks, range of motion exercises and peroneal muscle strengthening exercises were started. Full weight bearing with an air-stirrup brace was allowed from 4 weeks after surgery.

### 2.3. Statistical Analysis

The statistical analysis was performed using IBM SPSS version 23 (IBM Corp., Armonk, NY, USA). Normality was tested for all dependent variables using the Kolmogorov–Smirnov and Shapiro–Wilk methods, and both methods had *p*-values > 0.05. The analysis was conducted using parametric tests because all variables were normally distributed. A paired *t*-test was used to compare pre- and postoperative VAS and AOFAS scores. Statistical significance was accepted for *p*-values < 0.05.

## 3. Results

All 40 patients had a history of one or more sprains; 21 patients had a history of three or more sprains. The average time from initial ankle sprain to visiting a hospital was 6.6 years (range, from 1 month to 40 years). At presentation, the main complaints were ankle instability and pain (15 patients), pain only (15 patients), instability (7 patients), and cosmetic problems (3 patients).

Eleven patients had a cyst with an open skin wound (Figure 2), and eight of these patients had undergone surgery under a misdiagnosis of bursitis. Cysts were located anterior to the lateral malleolus in 22 cases, next to the lateral malleolus in 13 cases, posterior to the lateral malleolus in three cases, and across the entire lateral malleolus in two cases (Figure 3).

According to operative findings, the lateral joint capsule was ruptured in all cases, and a joint capsule defect was observed in 20 cases (Figure 4). Surgeries were performed in the following manner. In all cases, the ruptured joint capsule was repaired. In 20 cases, the repair site was reinforced using a fibular periosteal flap (Figure 5) and in one case using a peroneal fascia flap (Figure 6). Of the three patients with severe ankle instability, two patients underwent the Chrisman–Snook procedure and one patient underwent anatomical ligament reconstruction using an allograft tendon. Reverse sural artery flap was performed in two patients with severe soft tissue defects.

Mean VAS and AOFAS scores improved from 5.2 (range, from 1 to 7) and 72.3 (from 65 to 87) preoperatively to 1.1 (range, from 0 to 3) and 93.6 (range, from 85 to 100), respectively, at final follow-up visits. The comparison of VAS and AOFAS scores of preoperative and final follow-up using paired *t*-test are described in Table 1. In one case, the cyst recurred at the site of the previous cyst after an ankle sprain at 14 months after surgery. Repeat surgery confirmed that the joint capsule had ruptured at the fistula repair site. The ruptured joint capsule was repaired and reinforced using a fibular periosteal flap, and since, there has been no evidence of recurrence.

## 4. Discussion

This study has several strengths. First, to the best of our knowledge, it is the first study to describe the clinical manifestations, diagnosis, treatment methods, and treatment results for synovial fistula of the ankle joint. Second, considering the low incidence of synovial fistula, the study was conducted on a large number of subjects. Finally, the follow-up period was long enough to assess the patients’ postoperative progress.

In general, synovial fistulae are caused by damage to the joint capsule due to trauma or surgery. Synovial herniation occurs by fat pad herniation and subsequent sinus track formation after joint capsule damage, and results in synovial fistula formation due to the continuous release of joint fluid caused by joint motion and muscle contracture. Synovial fistulae are rare in joints, and synovial fistulae that occur after arthroscopic surgery in the knee joint are most commonly reported [6]. A synovial fistula develops as a complication in 0.01–0.61% after arthroscopy [23]. However, these rates may be underestimations due to the benign clinical course of the disease. In addition, there have been reports of synovial fistulae caused by complications after arthroscopic rotator cuff repair during shoulder joint and wrist arthroscopy [10,11]. Synovial fistulae have also been reported to occur after surgery on trigger fingers [24]. Rasmussen and Hjorth Jensen reported synovial fistula development in one case after 105 ankle arthroscopic procedures [25]. However, the majority of reports are case reports, and thus, little is known about its precise incidence. In this study, all patients had a history of acute ankle sprain during the 3 months preceding cyst development around the lateral malleolus. Based on this observation, we believe that rupture of the joint capsule due to acute sprain can cause a synovial fistula, and that a cyst around the lateral malleolus after ankle sprain should raise suspicion.

Patients with a synovial fistula of the ankle joint have been reported to complain of pain, instability, cosmesis, and a chronic open wound. Namm et al. reported, for 15 synovial fistulae of the hand, that VAS improved from six preoperatively to one after surgery [13]. In our cohort, pain was the major complaint, and 30 cases (75%) of all cases that complained of pain experienced pain relief after surgery.

Analyses of previous studies have shown that patients often complain of a chronic open wound and a cyst around a fistula or exudates from its orifice [2,8,12,14,26,27,28]. Cysts associated with synovial fistula characteristically change in size due to joint motion, unlike those associated with bursitis. In these cases, simple resection may lead to a chronic wound because continuous exudates through the fistula prevent wound recovery. In the present study, eight patients (20%) were misdiagnosed with bursitis at other hospitals, and chronic open wounds developed due to persistent fluid leakage after surgery. Therefore, it is important to differentiate synovial fistula and bursitis-associated cysts, though this is often difficult to achieve based on appearance. A synovial fistula can be distinguished from bursitis by a large mass-affected area around the lateral malleolus and size changes caused by joint motion. However, the definitive diagnosis of a synovial fistula requires confirmation of the presence of a fistula tract. Therefore, an arthrogram is helpful for confirming fistulous communication between the joint and synovial space (Figure 7). In addition, saline loading of the ankle joint enables the tract to be located intraoperatively.

There is no universal treatment available for a synovial fistula. In cases with infection, surgical debridement is mandatory, whereas in aseptic cases, initial conservative treatment is recommended. The consensus for treatment is immobilization with the joint in extension and compression for 1–2 weeks [6,29] and surgical closure if immobilization fails to promote healing. Excision and direct closure are commonly performed because of its simplicity, but fistulae often recur. Excision and coverage with a regional myofascial or a free flap are indicated, especially in cases with a chronic open wound [4,29,30]. In the present study, simple repair was performed initially, and additional augmentation using a fibular periosteal or peroneal fascia flap was performed when tissue quality around a synovial fistula was poor. Fibular periosteum, which has a strength equivalent to anterior tibiofibular ligament (ATFL), has been used for the surgical treatment of chronic ankle instability [18]. Furthermore, its harvesting is a straightforward, safe procedure and requires no additional incision. In addition, peroneal fascia flaps have been used for chronic wound coverage [19,31,32]. These flaps have the advantages of being thin, pliable, resilient, easily and rapidly dissected, well vascularized (they provide durable skin grafts), and of providing a consistent proximal blood supply [19].

This study is limited by its retrospective nature and because two surgeons performed the surgeries, which introduced the risk of bias. However, before beginning this study, the two surgeons unified surgical methods, rehabilitations, and outcome evaluations, which we believe minimized the risk of bias. Furthermore, the retrospective nature of the study made it difficult to assess the incidence of synovial fistula of the ankle joint accurately. Therefore, we believe a randomized, prospective study is necessary.

## 5. Conclusions

Cyst occurrence due to a synovial fistula should be considered when treating cysts around the lateral malleolus. Fistula repair and reinforcement with fibular periosteum provides a good treatment option for cysts attributed to synovial fistula of the ankle that fail to respond to conservative treatment.

## Figures and Tables

**Figure 1 jcm-11-06215-f001:**
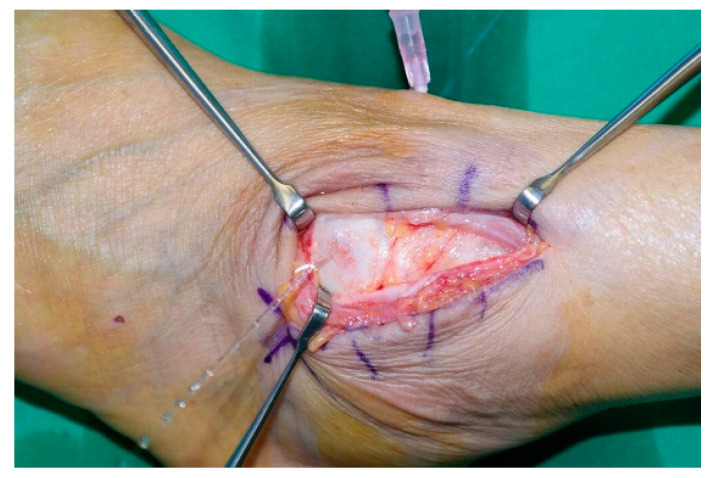
Saline leakage after saline loading into the ankle joint.

**Figure 2 jcm-11-06215-f002:**
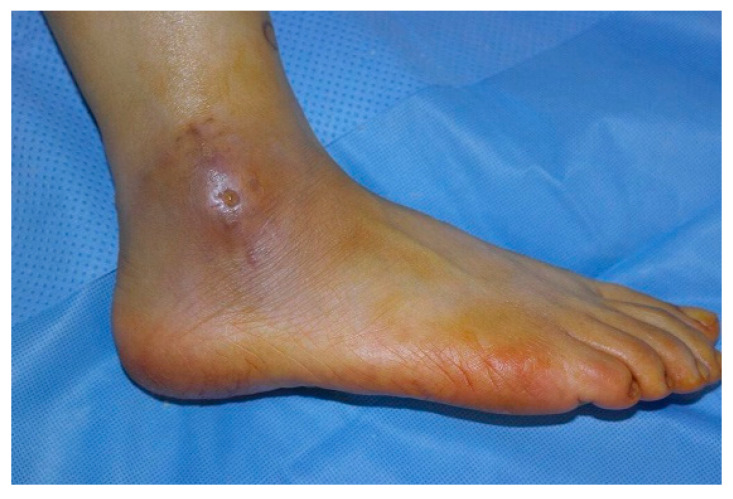
Cyst with an open skin wound.

**Figure 3 jcm-11-06215-f003:**
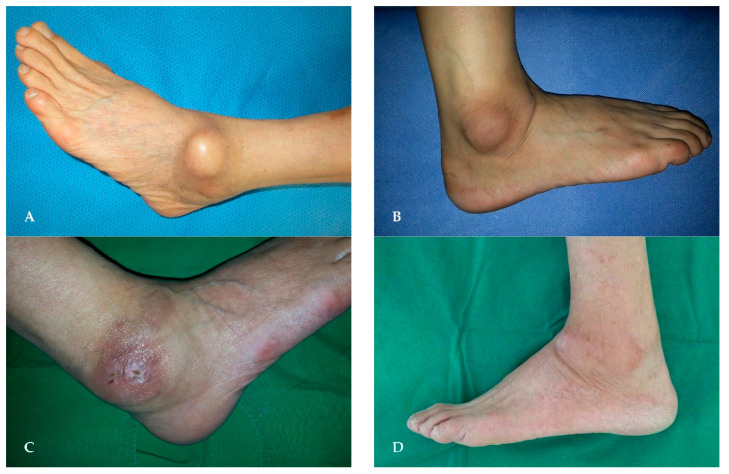
Cyst locations: (**A**) anterior to the lateral malleolus; (**B**) next to the lateral malleolus; (**C**) posterior to the lateral malleolus; (**D**) across the entire lateral malleolus.

**Figure 4 jcm-11-06215-f004:**
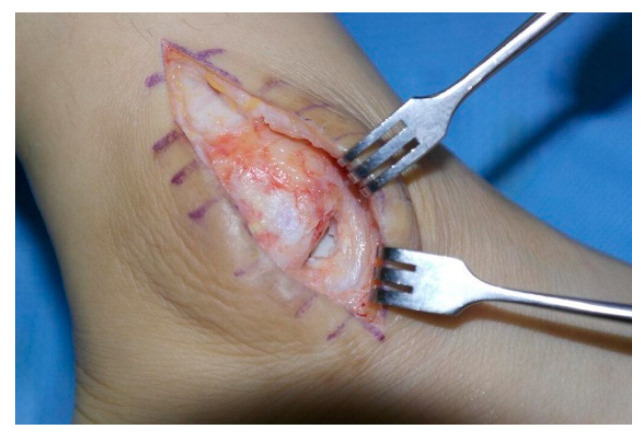
Defect of the ankle joint capsule.

**Figure 5 jcm-11-06215-f005:**
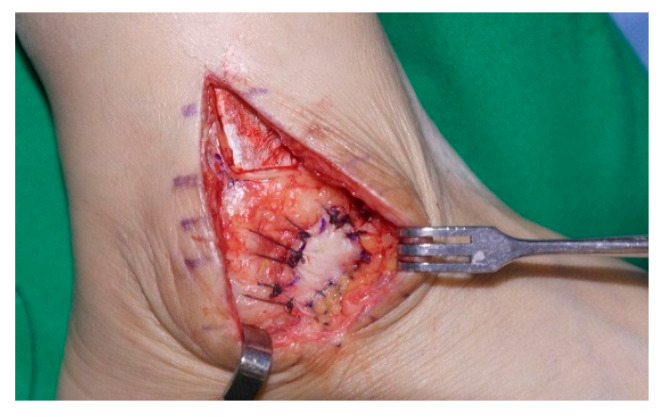
Joint capsule repaired and reinforced using a fibular periosteal flap.

**Figure 6 jcm-11-06215-f006:**
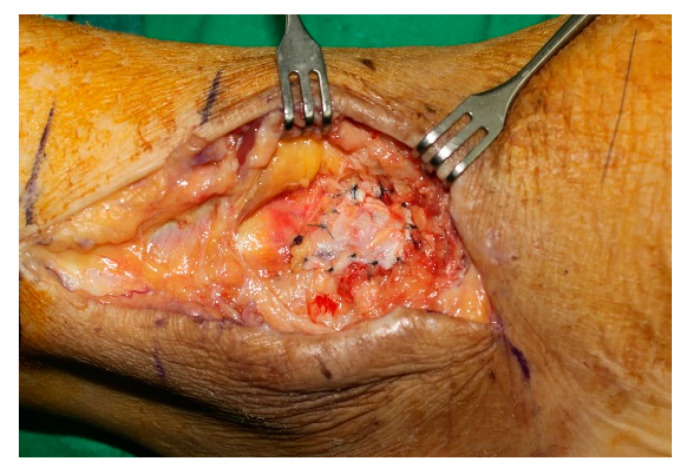
Joint capsule repaired and reinforced using a peroneal fascia flap.

**Figure 7 jcm-11-06215-f007:**
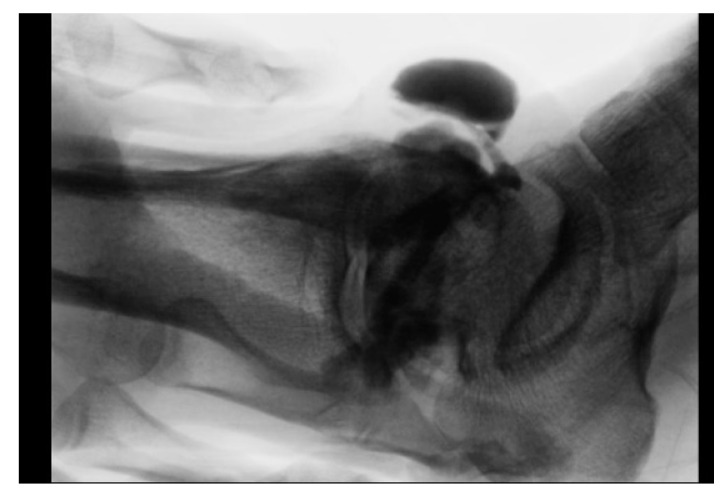
Arthrogram used to confirm fistulous communication between the joint and synovial space.

**Table 1 jcm-11-06215-t001:** The paired *t*-test of VAS and AOFAS scores.

Scores	Mean	SD	Effect Size (Cohen’s D)	95% Confidence Interval of the Difference		*p* Value
				Lower	Upper	
VAS	−4.1	0.96	2.86	−4.41	−3.79	<0.001
AOFAS	21.3	2.68	4.44	20.44	22.16	<0.001

Values are presented as number. VAS, visual analog scale; AOFAS, American Orthopedic Foot and Ankle Society; SD, standard deviation

## Data Availability

Data available on request due to restrictions of privacy.

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
