# Peer review of "Characteristics of Synovial Fistula of the Ankle Joint: A Case Series"

_jcm, 2022, doi:10.3390/jcm11206215_

Round 1

Reviewer 1 Report

Thank you for the opportunity to review your work. My biggest comments are on the statistical analysis. Please give the following questions:
1.    Give an analysis of the sample size -why was it decided to have 40 people?
2.    State the effect size of significant results.
3.    Add a confidence interval.
4.    Missing from the statistical analysis what test was used to determine whether the results have a normal distribution or deviate from the distribution?
5.    What program was used to calculate the statistics?
6.    In my opinion, the results are not very readable suggests: Pre- and post-operative result suggests to give additionally in a table in which the value of the test and the p-score, the effect size and the confidence interval.

Author Response

We sincerely thank you for reviewing this study.

  1. Give an analysis of the sample size -why was it decided to have 40 people?

: It was difficult to compare sample sizes because there were no previous studies on this topic. We tried to include as many patients as possible, and the sample size could not be calculated due to the nature of the retrospective study.

  1. State the effect size of significant results.

: The contents are added to the table 1.

  1. Add a confidence interval.

: The contents are added to the table 1.

  1. Missing from the statistical analysis what test was used to determine whether the results have a normal distribution or deviate from the distribution?

: Normaility was evaluated using the Kolmogorov-Smirnov & Shapiro-Wilk method, and both methods had a P-value of 0.05 or more. We have added content to the method.

  1. What program was used to calculate the statistics?

: We calculated the statistics using IBM SPSS version 23. We have added content to the method.

  1. In my opinion, the results are not very readable suggests: Pre- and post-operative result suggests to give additionally in a table in which the value of the test and the p-score, the effect size and the confidence interval.

: We have added content and table to the result.

Reviewer 2 Report

Review comments,

Thank you kindly for the opportunity to review the manuscript “Characteristics of Synovial Fistulae of the Ankle Joint“.

The present study is a case series reporting on the clinical manifestations, diagnosis, treatment methods, and treatment results for synovial fistula of the ankle joint. As synovial fistula is a rare condition, the outcomes from the present study should be valuable for physicians.

Thank you for the opportunity to review this manuscript.

TITLE

Please include the study design.

ABSTRACT

Well written. Easy to understand.

INTRODUCTION

Well written. Easy to understand.

MATERIALS AND METHOD

Appropriate. My concern is how you managed the patients with an open skin wounds. Is it the same as a patient without an open wound? Or did you perform culture tests, administer antimicrobials for a long period of time, check the inside of the joint with an arthroscope, etc.?

RESULTS

Concise and easy to understand

DICUSSIONCONCLUSION

Well written. Easy to understand.

Thank you very much for the opportunity to carry out this review.

Author Response

We sincerely thank you for taking your valuable time to review this study.

TITLE

Please include the study design.

: Study design has been added.

ABSTRACT

Well written. Easy to understand.

INTRODUCTION

Well written. Easy to understand.

MATERIALS AND METHOD

Appropriate. My concern is how you managed the patients with an open skin wounds. Is it the same as a patient without an open wound? Or did you perform culture tests, administer antimicrobials for a long period of time, check the inside of the joint with an arthroscope, etc.?

: Culture test was performed during surgery in all patients, including those with open skin wounds. If the bacteria were identified, an appropriate antibacterial agent was used.

Soft tissue defect was the biggest concern on the patients with an open skin wound. In our study, except for 2 patients who underwent reverse sural artery flap, primary closure was possible.

RESULTS

Concise and easy to understand

DICUSSION、CONCLUSION

Well written. Easy to understand.

Reviewer 3 Report

Chronic wounds around the foot and ankle are a significant cause of morbidity, and affect quality of life adversely. The authors deserve to be commended on this work. 

Specific comments:

Title should mention study design, i.e. case series

The statistical analysis section should specify VAS and AOFAS scores as the dependent variable. "clinical finding" is confusing.

Author Response

We sincerely thank you for reviewing this study.

  1. Title should mention study design, i.e. case series

: We added study design to the title.

  1. The statistical analysis section should specify VAS and AOFAS scores as the dependent variable. "clinical finding" is confusing

: We changed the “clinical finding” to a specific score.

Round 2

Reviewer 1 Report

Thank you for your reply. The corrections are acceptable.